# Unsupervised Image-to-Image Translation Using Domain-Specific Variational Information Bound

**Hadi Kazemi**
hakazemi@mix.wvu.edu

**Sobhan Soleymani**
ssoleyma@mix.wvu.edu

**Fariborz Taherkhani**
fariborztaherkhani@gmail.com

**Seyed Mehdi Iranmanesh**
seiranmanesh@mix.wvu.edu

**Nasser M. Nasrabadi**
nasser.nasrabadi@mail.wvu.edu
West Virginia University
Morgantown, WV 26505

## Abstract

Unsupervised image-to-image translation is a class of computer vision problems which aims at modeling conditional distribution of images in the target domain, given a set of unpaired images in the source and target domains. An image in the source domain might have multiple representations in the target domain. Therefore, ambiguity in modeling of the conditional distribution arises, specially when the images in the source and target domains come from different modalities. Current approaches mostly rely on simplifying assumptions to map both domains into a shared-latent space. Consequently, they are only able to model the domain-invariant information between the two modalities. These approaches usually fail to model domain-specific information which has no representation in the target domain. In this work, we propose an unsupervised image-to-image translation framework which maximizes a domain-specific variational information bound and learns the target domain-invariant representation of the two domain. The proposed framework makes it possible to map a single source image into multiple images in the target domain, utilizing several target domain-specific codes sampled randomly from the prior distribution, or extracted from reference images.

## 1 Introduction

Image-to-image translation is the major goal for many computer vision problems, such as sketch to photo-realistic image translation [25], style transfer [13], inpainting missing image regions [12], colorization of grayscale images [11, 32], and super-resolution [18]. If corresponding image pairs are available in both source and target domains, these problems can be studied in a supervised setting. For years, researchers [22] have made great efforts to solve this problem employing classical methods, such as superpixel-based segmentation [39]. More recently, frameworks such as conditional Generative Adversarial Networks (cGAN) [12], Style and Structure Generative Adversarial Network ($S^2$-GAN) [30], and VAE-GAN [17] are proposed to address the problem of supervised image-to-image translation. However, in many real-world applications, collecting paired training data is laborious and expensive [37]. Therefore, in many applications, there are only a few paired images available or no paired images at all. In this case, only independent sets of images in each domain,

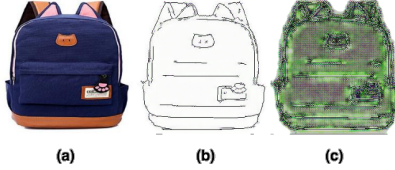

Figure 1: (a) The photo-realistic image. (b) Translated image in the edge domain, using CycleGAN. (c) Generated edges after Histogram Equalization to illustrate how photo-specific information are encoded to satisfy cycle consistency.

with no correspondence in the other domain, should be deployed to learn the cross-domain image translation task. Despite the difficulty of the unsupervised image-to-image translation, since there is no paired samples guiding how an image should be translated into a corresponding image in the other domain, it is still more desirable compared to the supervised setting due to the lack of paired images and the convenience of collecting two independent image sets. As a result, in this paper, we focus on the design of a framework for unsupervised image-to-image translation.

The key challenge in cross-domain image translation is learning the conditional distribution of images in the target domain. In the unsupervised setting, this conditional distribution should be learned using two independent image sets. Previous works in the literature mostly consider a shared-latent space, in which they assume that images from two domains can be mapped into a low-dimensional shared-latent space [37, 20]. However, this assumption does not hold when the two domains represent different modalities, since some information in one modality might have no representation in the other modality. For example, in the case of sketch to photo-realistic image translation, color and texture information have no interpretable meaning in the sketch domain. In other words, each sketch can be mapped into several photo-realistic images. Accordingly, learning a single domain-invariant latent space with aforementioned assumption [37, 20, 24] prevents the model from capturing domain-specific information. Therefore, a sketch can only be mapped into one of its corresponding photo-realistic images. In addition, since the current unsupervised techniques are implemented mainly based on the "cycle consistency" [20, 37], the translated image in the target domain may encode domain-specific information of the source domain (Figure 1). The encoded information can then be utilized to recover the source image again. This encoding can effectively degrade the performance and stability of the training process.

To address this problem, we remove the shared-latent space assumption, and learn a domain-specific space jointly with a domain-invariant space. Our proposed framework is based on Generative Adversarial Networks and Variational Autoencoders (VAEs), and models the conditional distribution of the target domain using VAE-GAN. Broadly speaking, two encoders map a source image into a pair of domain-invariant and source domain-specific codes. The domain-invariant code in combination with a target domain-specific code, sampled from a desired distribution, is fed to a generator which translates them into the corresponding target domain image. To reconstruct the source image at the end of the cycle, the extracted source domain-specific code is passed through a domain-specific path to the backward path from translated target domain image.

In order to learn two distinct codes for the shared and domain-specific information, we train the network to extract two distinct domain-specific and domain-invariant codes. The former is learned by maximizing its mutual information with the source domain while simultaneously we minimize the mutual information between this code and the translated image in the target domain. The mutual information maximization may also result in the domain-specific code to represent an interpretable representation of the domain-specific information [6]. These loss terms are crucial in the unsupervised framework, since domain-invariant information may also go through the domain-specific path to satisfy the cycle consistency in the backward path.

In this paper we extend CycleGAN [37] to learn a domain-specific code for each modality, through domain-specific variational information bound maximization, in addition to a domain-invariant code. Then, based on the proposed domain-specific learning scheme, we introduce a framework for one-to-many cross-domain image-to-image translation in an unsupervised setting.

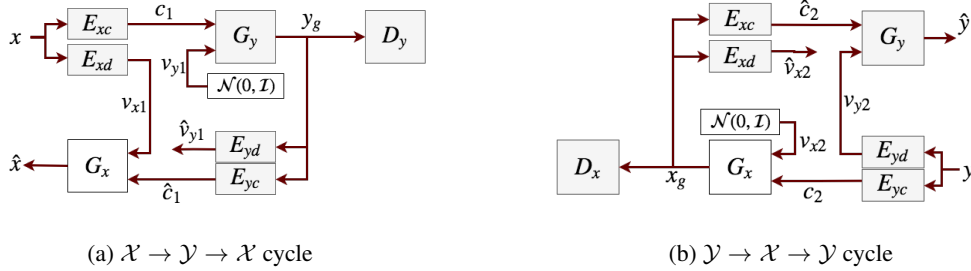

(a) $\mathcal{X} \rightarrow \mathcal{Y} \rightarrow \mathcal{X}$ cycle                    (b) $\mathcal{Y} \rightarrow \mathcal{X} \rightarrow \mathcal{Y}$ cycle

Figure 2: Proposed framework for unsupervised image-to-image translation.

## 2   Related Works

In the computer vision literature, image generation problem is tackled using autoregressive models [21, 29], restricted Boltzmann machines [26], and autoencoders [10]. Recently, generative techniques are proposed for image translation tasks. Models such as GANs [7, 34] and VAEs [23, 15] achieve impressive results in image generation. They are also utilized in conditional setting [12, 38] to address the image-to-image translation problem. However, in the prior research, relatively less attention is given to the unsupervised setting [20, 37, 4].

Many state-of-the-art unsupervised image-to-image translation frameworks are developed based on the cycle-consistency constraint [37]. Liu et al. [20] showed that learning a shared-latent space between the images in source and target domains implies the cycle-consistency. The cycle-consistency constraint assumes that the source image can be reconstructed from the generated image, in the target domain, without any extra domain-specific information [20, 37]. From our experience, this assumption severely constrains the network and degrades the performance and stability of the training process, in the case of learning the translation between different modalities. In addition, this assumption limits the diversity of generated images by the framework, i.e., the network associates a single target image with each source image. To tackle this problem, some prior research attempt to map a single image into multiple images in the target domain in a supervised setting [5, 3]. This problem is also addressed in [2] in an unsupervised setting. However, they have not considered any mechanisms to force their auxiliary latent variables to represent only the domain-specific information.

In this work, in contrast, we aim to learn distinct domain-specific and domain-invariant latent spaces in an unsupervised setting. The learned domain-specific code is supposed to represent the properties of the source image which have no representation in the target domain. To this end, we train our network by maximization of a domain-specific variational information to learn a domain-specific space.

## 3   Framework and Formulation

Our framework, as illustrated in Figure 2, is developed based on GAN [30] and VAE-GAN [17], and includes two generative adversarial networks; $\{G_x, D_x\}$ and $\{G_y, D_y\}$. The encoder-generators, $\{E_{xd}, G_x\}$ and $\{E_{yd}, G_y\}$, also constitute two VAEs. Inspired by CycleGAN model [37], we trained our network in two cycles; $\mathcal{X} \rightarrow \mathcal{Y} \rightarrow \mathcal{X}$ and $\mathcal{Y} \rightarrow \mathcal{X} \rightarrow \mathcal{Y}$, where $\mathcal{X}$ and $\mathcal{Y}$ represent the source and target domains, respectively.[1] Each cycle consists of forward and backward paths. In each forward path, we translate an image from the input domain into its corresponding image in the output domain. In the backward path, we remap the generated image into the input domain and reconstruct the input image. In our formulation, rather than learning a single shared-latent space between the two domains, we propose to decompose the latent code, $z$, into two parts: $c$, which is the domain-invariant code between the two domains, and $v_i, i = \{x, y\}$, which is the domain-specific code.

During the forward path in $\mathcal{X} \rightarrow \mathcal{Y} \rightarrow \mathcal{X}$ cycle, we simultaneously train two encoders, $E_{xc}$ and $E_{xd}$, to map data samples from the input domain, $\mathcal{X}$, into a latent representation, $z$. The input domain-invariant encoder, $E_{xc}$, maps the input image, $x \in \mathcal{X}$, into the input domain-invariant code, $c_1$. The input domain-specific encoder, $E_{xd}$, maps $x$ into the input domain-specific code, $v_{x1}$.

Then, the domain-invariant code, $c_1$, and a randomly sampled output domain-specific code, $v_{y1}$, are fed to the output generator (decoder), $G_y$, to generate the corresponding representation of the input image, $y_g = G_y(c_1, v_{y1})$, in the output domain $\mathcal{Y}$. Since in $\mathcal{X} \to \mathcal{Y} \to \mathcal{X}$ cycle the output domain-specific information is not available during the training phase, a prior, $p(v_y)$, is imposed over the domain-specific distribution which is selected as a unit normal distribution $\mathcal{N}(0, \mathcal{I})$. Here, index 1 in the codes' subscripts refers to the first cycle $\mathcal{X} \to \mathcal{Y} \to \mathcal{X}$. We use the same notation for all the latent codes in the reminder of the paper.

The output discriminator, $D_y$, is employed to enforce the translated images, $y_g$, resemble images in the output domain $\mathcal{Y}$. The translated images should not be distinguishable from the real samples in $\mathcal{Y}$. Therefore, we apply the adversarial loss [30] which is given by:

$$\mathcal{L}^1_{GAN} = \mathbb{E}_{y \sim p(y)} \log[D_y(y)] + \mathbb{E}_{(c_1, v_{y1}) \sim p(c_1, v_{y1})} \log[1 - D_y(G_y(c_1, v_{y1}))]. \quad (1)$$

Note that the domain-specific encoder $E_{xd}$ outputs mean and variance vectors $(\mu_{vx1}, \sigma^2_{vx1}) = E_{xd}(x)$, which represents the distribution of the domain-specific code $v_{x1}$ given by $q_x(v_{x1}|x) = \mathcal{N}(v_{x1}|\mu_{vx1}, diag(\sigma^2_{vx1}))$. Similar to the previous works on VAE [15], we assume that the domain-specific components of $v_x$ are conditionally independent and Gaussian with unit variance. We utilize reparametrization trick [15] to train the VAE-GAN using back-propagation. We define the variational loss for the domain-specific VAE as follows:

$$\mathcal{L}^1_{VAE} = -D_{KL}[q_x(v_{x1}|x), p(v_x)] + \mathbb{E}_{v_{x1} \sim q(v_{x1}|x)}[\log p(x|v_{x1})]. \quad (2)$$

where the Kullback–Leibler ($D_{KL}$) divergence term is a measure of how the distribution of domain-specific code, $v_x$, diverges from the prior distribution. The conditional distribution $p(x|v_{x1})$ is modeled as Laplacian distribution, and therefore, minimizing the negative log-likelihood term is equivalent to the absolute distance between the input and its reconstruction.

In the backward path, the output domain-invariant encoder, $E_{yc}$, and the output domain-specific encoder, $E_{yd}$, map the generated image into the reconstructed domain-invariant code, $\widehat{c_1}$, and the reconstructed domain-specific code, $\widehat{v_{y1}}$, respectively. The domain-specific encoder, $E_{yd}$, outputs mean and variance vectors $(\mu_{vy1}, \sigma^2_{vy1}) = E_{yd}(G_y(c_1, v_{y1}))$ which represents the distribution of the domain-specific code, $v_{y1}$, given by $q_y(v_{y1}|y) = \mathcal{N}(v_{y1}|\mu_{vy1}, diag(\sigma^2_{vy1}))$. Finally, the reconstructed input, $\widehat{x}$, is generated by the output generator, $G_x$, with $\widehat{c_1}$ and $v_{x1}$ as its inputs. Here, $v_{x1}$ is sampled from its distribution, $\mathcal{N}(\mu_{vx1}, diag(\sigma^2_{vx1}))$, where $(\mu_{vx1}, \sigma^2_{vx1})$ is the output of $E_{xd}$ in the forward path. We enforce a reconstruction criteria to force $\widehat{c_1}$, $\widehat{v_{y1}}$ and $\widehat{x}$ to be the reconstruction of $c_1$, $v_{y1}$, and $x$, respectively. To this end, the reconstruction loss is defined as follows:

$$\mathcal{L}^1_r = \mathbb{E}_{x \sim p(x), v_{y1} \sim \mathcal{N}(0, \mathcal{I})}[\lambda_1 ||\widehat{x} - x||_2 + \lambda_2 ||\widehat{v_{y1}} - v_{y1}||_2 + \lambda_3 ||\widehat{c_1} - c_1||_2], \quad (3)$$

where $\lambda_1$, $\lambda_2$, and $\lambda_3$ are the hyper-parameters to control the weight of each term in the loss function.

## 4 Domain-specific Variational Information bound

In the proposed model, we decompose the latent space, $z$, into the domain-invariant and domain-specific codes. As it is mentioned in the previous section, the domain-invariant code should only capture the information shared between the two modalities, while the domain-specific code represents the information which has no interpretation in the output domain. Otherwise, all the information can go through the domain-specific path and satisfy the cycle-consistency property of the network ($\mathbb{E}_{x \sim p(x)} ||\widehat{x} - x||_2 \to 0$ and $\mathbb{E}_{y \sim p(y)} ||\widehat{y} - y||_2 \to 0$). In this trivial solution, the generator, $G_y$, can translate an input domain image into the output domain image that does not correspond to the input image, while satisfying the discriminator $D_y$ in terms of resembling the images in $\mathcal{Y}$. Figure 7 (second row) presents images generated by this trivial solution.

Here, we propose an unsupervised method to learn the domain-specific information of the source data distribution which has minimum information about the target domain. To learn the source domain-specific code, $v_x$, we propose to minimize the mutual information between $v_x$ and the target domain distribution, while simultaneously, we maximize the mutual information between $v_x$ and the source domain distribution. Similarly, the target domain-specific code $v_y$ is learned for target domain $\mathcal{Y}$. In other words, to learn the source and target domain specific codes $v_x$ and $v_y$, we should minimize the following loss function:

$$\mathcal{L}_{int} = \big(I(y, v_x; \theta) - \beta I(x, v_x; \theta)\big) + \big(I(x, v_y; \theta) - \beta I(y, v_y; \theta)\big), \quad (4)$$

where $\theta$ represents the model parameters. To translate $\mathcal{L}_{int}$ to an implementable loss function, we define the following two loss functions:

$$\mathcal{L}_{int}^1 = I(x, \widehat{v}_{y_1}; \theta) - \beta I(x, v_{x_1}; \theta), \qquad \mathcal{L}_{int}^2 = I(y, \widehat{v}_{x_2}; \theta) - \beta I(y, v_{y_2}; \theta), \qquad (5)$$

where $\mathcal{L}_{int}^1$ and $\mathcal{L}_{int}^2$ are implemented in cycles $\mathcal{X} \to \mathcal{Y} \to \mathcal{X}$ and $\mathcal{Y} \to \mathcal{X} \to \mathcal{Y}$, respectively.

Instead of minimizing $\mathcal{L}_{int}^1$, or similarly $\mathcal{L}_{int}^2$, we minimize their variational upper bounds, which we refer to as domain-specific variational information bounds. Zhao et al. [35] illustrated that using KL-divergence in VAEs results in *information preference* problem, in which the mutual information between the latent code and the input becomes vanishingly small, while training the network using only reconstruction loss, with no KL divergence term, maximizes the mutual information. However, some other types of divergences, such as MMD and Stein Variational Gradient, do not suffer from this problem. Consequently, in this paper, for $\mathcal{L}_{int}^1$, to maximize $I(x, v_{x_1}; \theta)$ we can replace the first term in (2) with Maximum-Mean Discrepancy (MMD) [35], which always prefers to maximize mutual information between $x$ and $v_{x_1}$. The MMD is a framework which utilizes all of the moments to quantify the distance between two distributions. It could be implemented using the kernel trick as follows:

$$MMD[p(z) \parallel q(z)] = E_{p(z),p(z')}[k(z, z')] + E_{q(z),q(z')}[k(z, z')] - 2E_{p(z),q(z')}[k(z, z')], \quad (6)$$

where $k(z, z')$ is any universal positive definite kernel, such as Gaussian $k(z, z') = e^{-\frac{\|z - z'\|}{2\sigma^2}}$. Consequently, we rewrite the VAE objective in Equation (2) as follows:

$$\mathcal{L}_{VAE}^1 = MMD[p(v_{x_1}) \parallel q(v_{x_1})] + \mathbb{E}_{v_{x1} \sim q(v_{x1}|x)}[\log p(x|v_{x1})]. \qquad (7)$$

Following the method described in [1], to minimize the first term of $\mathcal{L}_{int}^1$ in (5), we define an upper-bound for the first term as:

$$I(x, \widehat{v}_{y_1}; \theta) \leq \int d\widehat{v}_{y_1} dx p(x) p(\widehat{v}_{y_1}|x) \log \frac{p(\widehat{v}_{y_1}|x)}{r(\widehat{v}_{y_1})} = L^1. \qquad (8)$$

Since $p(\widehat{v}_{y_1})$ is tractable but difficult to compute, we define variational approximations to it as $r(\widehat{v}_{y_1})$. Similar to [1], $r(z)$ is defined as a fixed $dim$-dimensional spherical Gaussian, $r(z) = N(z|0, I)$, where $dim$ is the dimension of $v_{y1}$. This upper-bound in combination with the MMD forms a domain-specific variational information bound. Note that MMD does not optimize an upper-bound to the negative log likelihood directly, but it guarantees the mutual information to be maximized and we can expect a high log likelihood performance [35]. To translate this upper-bound, $L^1$, to an implementable loss function in the model, we use the following empirical data distribution approximation:

$$p(x) \approx \frac{1}{N} \sum_{n=1}^N \delta_{x_n}(x). \qquad (9)$$

Therefore, the upper bound can be approximated as:

$$L^1 \approx \frac{1}{N} \sum_{n=1}^N \int d\widehat{v}_{y_1} p(\widehat{v}_{y_1}|x_n) \log \frac{p(\widehat{v}_{y_1}|x_n)}{r(\widehat{v}_{y_1})}. \qquad (10)$$

Since $\widehat{v}_{y_1} = f(x, v_{y_1})$ and $v_{y_1} \sim \mathcal{N}(0, \mathcal{I})$, the implementable upper-bound, $L$, can be approximated as follows:

$$L^1 \approx \frac{1}{N} \sum_{n=1}^N \mathbb{E}_{v_{y_1} \sim \mathcal{N}(0,\mathcal{I})} D_{KL}[p(\widehat{v}_{y_1}|x_n) \| r(\widehat{v}_{y_1})]. \qquad (11)$$

As illustrated in Figure 2b, we train the $\mathcal{Y} \to \mathcal{X} \to \mathcal{Y}$ cycle starting from an image $y \in \mathcal{Y}$. All the components in this cycle share weights with the corresponding components in $\mathcal{X} \to \mathcal{Y} \to \mathcal{X}$ cycle. Similar losses, $L^2$, $\mathcal{L}_r^2$, $\mathcal{L}_{VAE}^2$, and $\mathcal{L}_{GAN}^2$, can be defined for this cycle. The overall loss for the network is defined as:

$$\text{Loss} = \sum_{i=1}^2 \alpha_1^i L^i + \alpha_2^i \mathcal{L}_r^i + \alpha_3^i \mathcal{L}_{GAN}^i + \alpha_4^i \mathcal{L}_{VAE}^i. \qquad (12)$$

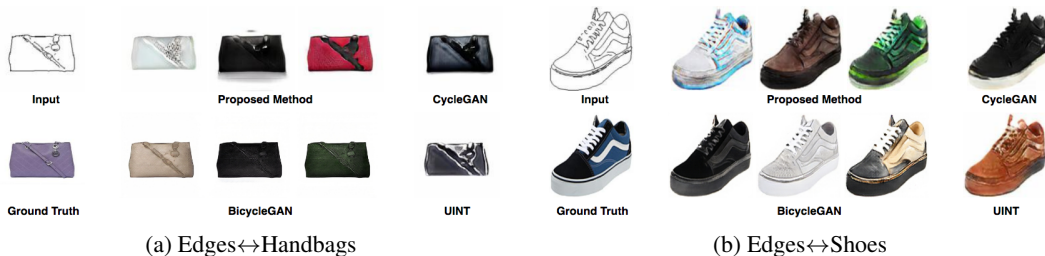

| (a) Edges↔Handbags | (b) Edges↔Shoes |

Figure 3: Qualitative comparison of our proposed method with BicycleGAN, CycleGAN and UNIT. The proposed framework is able to generate diverse realistic outputs. However, it does not require any supervisions during its training phase.

## 5 Implementation

We adopt the architecture for our common latent encoder, generator, and discriminator networks from Zhu and Park et al. [37]. The domain-invariant encoders includes two stride-2 convolutions, and three residual blocks [8]. The generators consist of three residual blocks and two transposed convolutions with stride-2. The domain-specific encoders share the first two convolution layers with their corresponding domain-invariant encoders, followed by five stride-2 convolutions. Since the spatial size of the domain-specific codes do not match with their corresponding domain-invariant codes, we tile them to the same size as the domain-invariant codes, and then, concatenate them to create the generators' inputs. For the discriminator networks we use $30 \times 30$ PatchGAN networks [19, 12], which classifies whether $30 \times 30$ overlapping image patches are real or fake. We use Adam optimizer [14] for online optimization with the learning rate of 0.0002. For reconstruction loss in (3), we set $\lambda_1 = 10$ and $\lambda_2 = \lambda_3 = 1$. The values of $\alpha_2$ and $\alpha_3$ in (12) are set to 1, and the $\frac{\alpha_4}{\alpha_1} = \beta = 1$. Finally, regarding the kernel parameter $\sigma$ in (6), as discussed in [35], MMD is fairly robust to this parameter selection, and using $\frac{2}{dim}$ is a practical value in most scenarios, where $dim$ is the dimension of $v_{x1}$.

## 6 Experiments

Our experiments aim to show that an interpretable representation can be learned by the domain-specific variational information bound maximization. Visual results on translation task show how domain-specific code can alter the style of generated images in a new domain. We compare our method against baselines both qualitatively and quantitatively.

### 6.1 Qualitative Evaluation

We use two datasets for qualitative comparison, edges ↔ handbags [36] and edges ↔ shoes [31]. Figures 3a and 3b represent the comparison between the proposed framework and baseline image-to-image translation algorithms: CycleGAN [37], UNIT [20], and BicycleGAN [38]. Our framework, similar to the BicycleGAN, can be utilized to generate multiple realistic images for a single input, while does not require any supervision. In contrast, CycleGAN and UNIT learn one-to-one mappings as they learn only one domain-invariant latent code between the two modalities. From our experience, training CycleGAN and UNIT on edges ↔ photos datasets is very unstable and sensitive to the parameters. Figure 1 illustrates how CycleGAN encodes information about textures and colors in the generated image in the edge domain. This information encoding enables the discriminator to easily distinguish the fake generated samples from the real ones which results in unstability in the training of the generators.

Three other datasets, namely architectural labels ↔ photos from the CMP Facade database [28], and CUHK Face Sketch Dataset (CUFS) [27] are employed for more qualitative evaluation. The image-to-image translation results for the proposed framework are presented in Figure 4d, and 4c for these datasets, respectively. Our method successfully captures domain-specific properties of the target domain. Therefore, we are able to generate diverse images from a single input sample. More results for edges ↔ shoes and edges ↔ handbags datasets are presented in Figures 4a and 4b, respectively. These figures present one-to-many image translation when different domain-specific

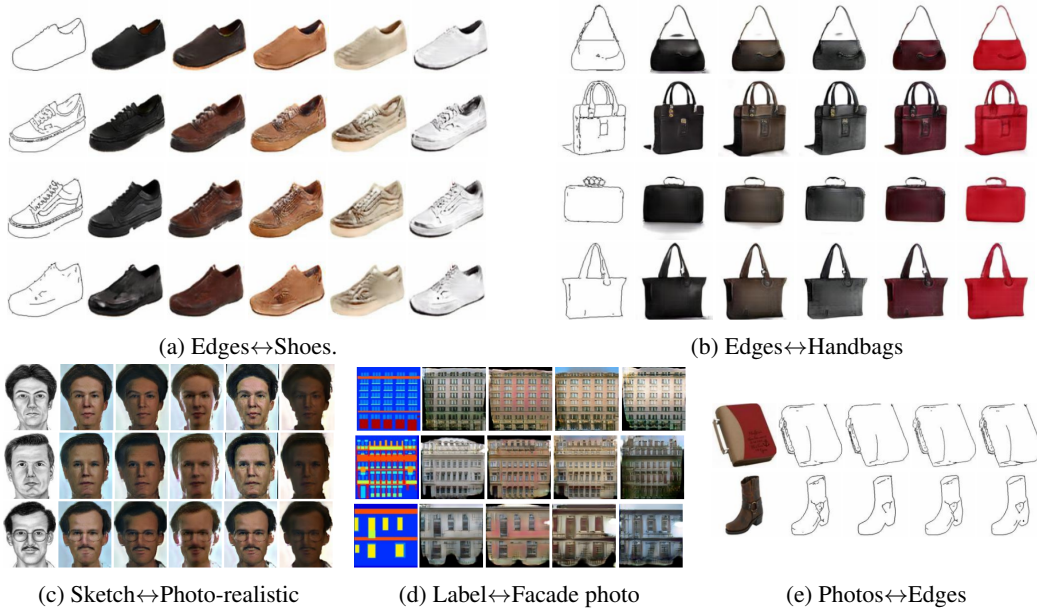

(a) Edges↔Shoes.

(b) Edges↔Handbags

(c) Sketch↔Photo-realistic

(d) Label↔Facade photo

(e) Photos↔Edges

Figure 4: The results of our framework on different datasets.

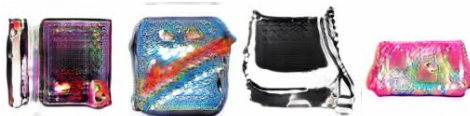

Figure 5: Failure cases, where some domain-specific codes do not result in well-defined styles.

codes are deployed. The results for the backward path for edges ↔ handbags and edges ↔ shoes are also presented in Figure 4e. Since there is no extra information in the edge domain, the generated edges are quite similar to each other despite the value of edge domain-specific code.

Using the learned domain-specific code, we can transfer domain-specific properties from a reference image in the output domain to the generated image. To this end, instead of sampling from the distribution of output domain-specific code, we can use a domain-specific code extracted from a reference image in the output domain. To this end, the reference image is fed to the output domain-specific encoder to extract its domain-specific code. The extracted code can be used for image translation guided by the reference image. Figures 6 show the results using domain-specific codes extracted from multiple reference images to translate edges into realistic photos. Finally, Figure 5 illustrates some failure cases, where some domain-specific codes do not result in well-defined styles.

## 6.2 Quantitative Evaluation

Table 1 presents the quantitative comparison between the proposed framework and three state-of-the-art models. Similar to BicycleGAN [38], we perform a quantitative analysis of the diversity using Learned Perceptual Image Patch Similarity (LPIPS) metric [33]. The LPIPS distance is calculated as the average distance between 2000 pairs of randomly generated output images, in deep feature space of a pre-trained AlexNet [16]. Diversity scores for different techniques using the LPIPS metric are summarized in Table 1. Note that the diversity score is not defined for one-to-one frameworks, e.g., CycleGAN and UNIT. Previous findings showed that these models are not able to generate large output variation, even by noise injection [12, 38]. The diversity scores of our proposed framework are close to the BicycleGAN, while we do not have any supervision during the training phase.

Generating unnatural images usually results in a high diversity score. Therefore, to investigate whether the variation of generated images is meaningful, we need to evaluate the visual realism of the generated samples as well. As proposed in [32, 37], the "fooling" rate of human subjects,

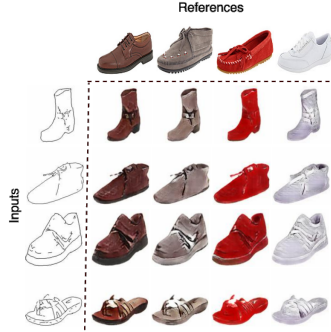

Figure 6: Using domain-specific information from a reference image to transform an input image into the output domain.

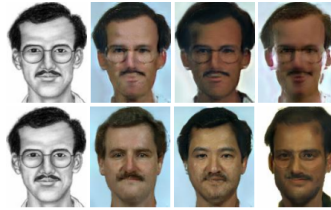

Figure 7: Generated images with (first row) and without (second row) mutual information minization between the target domain-specific code and the source domain.

Table 1: Diversity measure for generated images using average LPIPS distance and realism score using human fooling rate, and FID score on Edges↔Shoes and edges ↔ handbags tasks.

| Method | Edges↔Shoes | | | Edges↔Handbags | | |
|---|---|---|---|---|---|---|
| | LPIPS Distance | Fooling Rate | FID Score | LPIPS Distance | Fooling Rate | FID Score |
| Real Images | 0.290 | - | - | 0.369 | - | - |
| UNIT | - | 22.0 | 90.32 | - | 19.2 | 84.36 |
| CycleGAN | - | 24.3 | 86.54 | - | 25.9 | 81.22 |
| BicycleGAN | 0.113 | 38.0 | 43.18 | 0.134 | 34.9 | 37.79 |
| Ours | 0.121 | 36.0 | 48.36 | 0.129 | 33.2 | 40.84 |

is considered as visual realism score of each framework. We sequentially presented a real and generated image to a human for 1 second each, in a random order, asked them to identify the fake, and measured the *fooling* rate. We also used the Frechet Inception Distance (FID) to evaluate the quality of generated images [9]. It directly measures the distance between the synthetic data distribution and the real data distribution. To calculate FID, images are encoded with visual features from a pre-trained inception model. Note that a lower FID value interprets as a lower distance between synthetic and real data distributions. Table 1 shows how the FID results confirm the results from fooling rate. We calculate the FID over 10k randomly generated samples.

### 6.3 Discussion and Ablation Study

Our framework learns a disentangled representation of content and style, which provides users more control on the image translation outputs. This framework is not only suitable for image-to-image translation, but also one can use it to transfer style between the images of a single domain. Comparing with other unsupervised one-to-one image-to-image translation frameworks, i.e., CycleGAN and UNIT, our method handles translation between significantly different domains. In contrast, CycleGAN encodes the domain-specific codes to satisfy the cycle-consistency (see Figure 1). UNIT also completely fails as it cannot find a shared representation in these cases.

Neglecting the minimization of the mutual information between target domain-specific information and the source domain may result in capturing attributes with high variation in the target despite their common nature in both domains. For example, as illustrated in Figure 7, the domain-specific code can result in altering the attributes, such as gender or face structure, while these attributes are domain-invariant properties of the two modalities. In addition, removing the domain-specific code cycle-consistency criteria (e.g. $v_{y1} = \hat{v}_{y1}$) results in a partial mode collapse in the model, with many outputs being almost identical, which reduces the LPIPS (see Table 2). Without the domain-invariant code cycle-consistency criteria (e.g. $c_1 = \hat{c}_1$), the image quality is unsatisfactory. A possible reason for quality degradation is that $c_1$ can include the domain-specific information as there is no constraint on it to represent shared information exclusively. That results in the same issue as explained in

Table 2: Average LPIPS distances with and without domain-specific code cycle-consistency on Edges↔Shoes and edges ↔ handbags tasks.

|  | shoes | | handbags | |
| --- | --- | --- | --- | --- |
|  | w/ | w/o | w/ | w/o |
| LPIPS | 0.121 | 0.095 | 0.129 | 0.113 |

Figure 1. Very small values for $\beta$ result in the second term in $\mathcal{L}_{int}^1$ in (5) to be neglected. Therefore, the domain-specific code, $v_{x1}$, will be irrelevant in the loss minimization and the learned domain specific code could be meaningless. In contrast, with very large values of $\beta$, $y_g$ carries the domain specific information of the $x$ as well.

## 7  Conclusion

In this paper, we introduced a framework for one-to-many cross-domain image-to-image translation in an unsupervised setting. In contrast to the previous works, our approach learns a distinct domain-specific code for each of the two modalities, maximizing a domain-specific variational information bound. In addition, it learns a domain-invariant code. During the training phase, a unit normal distribution is imposed over the domain-specific latent distribution, which let us control the domain-specific properties of the generated image in the output domain. To generate diverse target domain images, we extract domain-specific codes from reference images, or sample them from a prior distribution. These domain-specific codes, combined with the learned domain-invariant code, result in target domain images with different target domain-specific properties.

## Footnotes

[1]For simplicity, in the remainder of the paper, for each cycle, we use terms input domain and output domain.

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
