[Reviews · NeurIPS 2018]

Reviewer 1



Update: overall I'm quite happy with the response and I think the work is great overall. However, the writing needs to be cleared up based on all the feedback to make this a better paper. Based on the extra information in the response I am updating my score to a 6. === SYNOPSIS: This paper proposes an unsupervised method based on GANs and Variational Auto-endcoders (VAEs) for performing one-to-many cross-domain image-to-image translation by optimizing a variational information bound during training. Instead of mapping both domains into a shared latent space, the authors separate the space into a domain-specific and domain-invariant space. By implementing the generator as a VAE, the method allows for sampling from the target side. HIGH-LEVEL FEEDBACK: The overall idea is neat and in line with several recent works in the Domain Adaptation and Image-to-Image Translation literature which either i) separate the latent space into learned private and shared spaces, or ii) employ VAEs. However, the combination of these two into one framework is novel to the best of my knowledge. The model description is thorough, but I do think the authors are unnecessarily verbose at times, and can improve the paper's overall readability. I had to read and re-read several sections to get the gist of what was done. The Experimental section is the weakest in my opinion. As it is known to be hard to do quantitative experiments with these models, I was at least looking for a more detailed qualitative analysis and discussion. Instead of what appears to be cherry-picked generations, it would also be instructive to view the failure cases as well? In particular, the Discussion section seems to be almost completely missing? Furthermore, there are no ablation results. Eg, what happens as you vary the \beta's in Eqn 4 or the \alpha's in Eqn 12? Note: The recent "XGAN" paper is very related, and tries to solve a very similar problem, and might be good to compare to [https://arxiv.org/pdf/1711.05139.pdf] OVERALL The model is interesting, and the results presented look promising. However, I think the paper needs a bit more work, and especially the Experimental section needs to be expanded to include more analyses of failure cases and ablation studies to get the paper up to NIPS standard. Fig 2 caption: "inforamtion"->"information"

Reviewer 2



This paper extend the CycleGAN by spliting the latent code into content and domain parts and achieves encouraging results in one-to-many unsupervised image-to-image translation. The entire framework is clean and well motivated. The paper is clearly written and easy to follow. It would be better to evaluate the chosen of hyper-parameters and show the influence of hyper-parameters on the performance.

Reviewer 3



Summary: The main contribution of this paper is a new framework for unsupervised image-to-image translation. Specifically, instead of learning only the domain-invariant information between two domains, the proposed approach extends CycleGAN to learn a domain-specific code for each modality and a domain-invariant code shared by the two modalities. The domain-specific code is learned via a domain-specific variational information bound. The trained model can be used for unsupervised one-to-many image-to-image translation. Strengths: (1) Significance: Though unsupervised image-to-image translation has been studied before, this paper pointed out one common drawback of such methods, i.e., no domain-specific code is learned for each individual modality. I think this is a good point, and the proposed method nicely solves this problem, and thus has some significance. The topic is also suitable for general NIPS readers. (2) Originality: The proposed model is interesting and seems novel. I like the idea of decomposing the latent code into the domain-invariant and domain-specific parts. The idea of using a domain-specific variational information bound to enforce the model disentangle the latent code also seems interesting. Weaknesses: (1) Clarity: I like the first three sections, which introduces the problem and proposed approaches clearly. However, Section 4 is generally not well written, and not easy to follow. Please see the Questions section for more details. For presentation, I think the introduction part talks about a little bit too many details. For example, it is not clear to me why putting Figure 1 & 2 in Section 1, and it seems to me moving these two figures to the experimental section and discussing the figures there might be better. Also, in Table 1, the authors showed the human evaluation results. However, how this human evaluation is performed is not discussed. I would recommend describing this in the main text. (2) Quality: I think the idea is novel, but the authors did not introduce the proposed variational information bound in Section 4 clearly, which seems to be a key contribution of this paper. Also, the authors did not provide enough experimental details for other people to reproduce the experimental results. Specifically, I have the following questions. Questions: (1) It is not clear to me why changing from Eqn. (2) to Eqn. (7), i.e., replacing KL with MMD will maximize the mutual information between x and v_{x1}. Please explain it clearly. I do not get the argument stated in Lines 166 to 169. (2) I understand that the bound in Eqn. (8) is to make the mutual information trackable. However, the formulation of the variational approximation r(\hat{v}_{y1}) is never introduced. Please clarify this. (3) There are just too many hyper-parameters in the final proposed method. For example, \lambda_{1,2,3}, \beta, \alpha_{1,2,3,4}^{1,2}, the bandwidth value \sigma in the Gaussian kernel of MMD. I'd like to know how these hyperparameters are selected, how sensitive these numbers are, and what are the numbers being used in the experiments. For the current draft, I think it will be difficult to reproduce the experiments. Overall, I think this paper is marginally below the acceptance threshold. Though the idea is novel, the presentation is not satisfying. If Section 4 is described more clearly, it has a chance to be a good paper. Minor issues: (1) Line 13: "domain" => "domains", Line 77: "Works" => "Work" (2) Equation (2) & (7): the second expectation term: "v_{x1} \sim p(v_{x1}|x)" => "v_{x1} \sim q_x(v_{x1}|x)" (3) It seems to me Figure 5 & 6 demonstrates the same thing. I am not sure why plotting these two figures separately here. I suggest combining these two figures, and getting rid of one of Figure 5(b) and Figure 6(c). These two subfigures basically are the same. -- After Rebuttal -- I have read the rebuttal and other reviewers' comments. Generally, I think the authors partially solved my concerns. For example, they discussed how they select hyper-parameters, and more technical details of the method are provided. As to quantitative analysis, it seems using LPIPS and fooling rate aligns well with the previous work. I feel this paper has some novelty inside, and in the camera-ready version, as the authors claim, I hope the authors will take the comments into consideration to make the presentation more clear. I slightly prefer to voting acceptance, and therefore moving the score to a 6.